# Quality of life and associated factors among patients with epilepsy at specialized hospitals, Northwest Ethiopia; 2019

**Fentahun Minwuyelet[1], Henok Mulugeta[1], Dejene Tsegaye[1], Benalfew lake[1], Asmare Getie** [2]*, **Baye Tsegaye[3], Getachew Mullu[4]**

**1** Department of Nursing, College of Health Science, Debre Markos University, Debre Markos, Ethiopia, **2** School of Nursing, Arba Minch University, Arba Minch, Ethiopia, **3** Department of Nursing, College of Medicine and Health Sciences, Wolkite University, Wolkite, Ethiopia, **4** Department of Midwifery, College of Health Science, Debre Markos University, Debre Markos, Ethiopia

* asmaregetie2017@gmail.com

**Data Availability Statement:** All relevant data are within the manuscript and its Supporting information files.

## Abstract

### Background

Epilepsy is a chronic non-communicable brain disorder and the second most burdensome disease in terms of disability-adjusted life years which is characterized by recurrent epileptic seizures, and a constant threat to the quality of life of the patient. Nearly 80% of people with epilepsy live in low- and middle-income countries and the risk of premature death in people with epilepsy is up to three times higher than for the general population. In many parts of the world, people with epilepsy and their families suffer from stigma and discrimination. This study was aimed to assess the quality of life and associated factors among adult people living with epilepsy using the world health organization quality of life assessment tool.

### Methods

Institution-based cross-sectional study design was conducted on 419 epileptic patients using a systematic random sampling technique. The data were collected using the WHO-QOL-BREF questionnaire. The data were entered into EpiData version 3.1 and exported to SPSS version 25 software for further analysis and bivariate and multivariable binary logistic regression analyses were done to identify factors associated with the dependent variable. The level of significance was declared as P value <0.05.

### Result

A total of 402 epileptic patients with a median age of 28 years were involved in the study. The result of this study was revealed that 47.8% (95% CI: 42%, 52%) of the respondents had poor quality of life. Respondents who were in the middle age group (AOR = 0.36, 95% CI: 0.19, 0.70), lower educational level (AOR = 3.11, 95%CI: 1.35, 7.15), those who had low drug adherence (AOR = 8.36, 95%CI: (2.41, 28.98) comorbid anxiety, (AOR = 3.63, 95% CI: 2.55, 8.42) and depression (AOR = 3.85, 95% CI: 2.16, 6.82) were found to be significantly associated with poor quality of life of epilepsy patients.

**Funding:** The author(s) received no specific funding for this work.

**Competing interests:** The authors have declared that no competing interests exist.

**Abbreviations:** AED, Adherence to an epileptic drug; AOR, adjusted odds ratio; COR, crude odds ratio; IRB, Institutional Review Board; PWE, patient with epilepsy; QOL, quality of life.

## Conclusion

This study revealed that almost one in two epileptic patients had poor quality of life. Age of the respondents, lower educational level, comorbid anxiety and depression, and lower adherence to drugs were significantly associated with poor quality of life. Therefore, health institutions and clinicians should not focus only on the treatment of the disease itself rather they should address diseases' impact on the quality of life of patients.

## Introduction

Epilepsy is a chronic non-communicable brain disorder characterized by recurrent epileptic seizures due to sudden abnormal excessive discharge of the cerebral neurons or brain cells [1]. Individuals who have had only febrile seizures and people with acute symptomatic seizures secondary to other disease is not epilepsy [2, 3]. Seizure is usually of brief duration and may produce post-seizure impairment and has brief periods of disruption, which include phenomena such as bodily distortion, loss of consciousness, injuries and its recurrence is a constant threat to the quality of life of the patient with epilepsy [4, 5].

Globally, an estimated five million people are diagnosed with epilepsy each year. In high-income countries, there are estimated to be 49 per 100 000 people diagnosed with epilepsy each year. In low- and middle-income countries, this figure can be as high as 139 per 100 000. This is likely due to the increased risk of endemic conditions such as malaria, the higher incidence of road traffic injuries; birth-related injuries; and variations in medical infrastructure, the availability of preventive health programs, and accessible care. Nearly 80% of people with epilepsy live in low- and middle-income countries and the risk of premature death in people with epilepsy is up to three times higher than for the general population. In many parts of the world, people with epilepsy and their families suffer from stigma and discrimination [5–7].

In Ethiopia, epilepsy is a public health problem, with an estimated prevalence of the disease in the country reported being 5.2/1000 inhabitants at risk with an annual incidence of 64 in100,000 inhabitants in large scale, rural, and community-based studies [3, 8].

Epilepsy has significant implications for patients' health and social functioning and has significantly higher rates of health-related quality of life impacts due to the burden of medication use, higher socioeconomic cost, lower employment rates, and lower income compared with healthy subjects [9]. People with epilepsy are subjected to social stigma which increases the risk of poor self-esteem, depression, anxiety, and suicide due to fear of having the next seizure which is reduced the quality of life of patients [10]. Therefore, it is important that health care providers routinely measure the impact of the complex pharmaco–psycho-social therapy given. Assessing the success of such a holistic method of care by determining the extent of seizure control with medication and monitoring for reduction of seizure frequency is important [11, 12].

Now a day quality of life (QOL) assessment has attracted more attention because it reveals complaints regarding attention, learning, physical pain, and health-related quality of life associated with epilepsy [13]. Increasingly, health care planners are recognizing that measures of disease alone are insufficient determinants of health status. Therefore; an individual's perception of their position in life in the context of the culture and value systems in which they live, and with their goals, expectations, standards, and concerns matter to assess the QOL of patients with epilepsy [14]. Epilepsy influences many dimensions of the quality of life of people with epilepsy than other chronic diseases both by the nature of the disorder and its associated

effects like problems in education, employment, marriage, perceived discrimination, comorbid anxiety and depression, and the outward manifestation of the symptoms [15, 16].

The World Health Organization (WHO's) 2010 Global Burden of Disease study ranks epilepsy as the second most burdensome neurologic disorder worldwide in terms of disability-adjusted life years [17]. Nearly 70 million people are suffering from epilepsies throughout the world. Epilepsy contributes a 1% burden to global diseases and out of this 80% is contributed in the developing countries [18]. The prevalence and incidence of epilepsy in different countries are varying. The overall prevalence of epilepsy in the worldwide is 10 per 1000 persons and the estimated prevalence of epilepsy in Africa was 7.85 per 1,000 persons [19–21].

Epilepsy highly affects the quality of life. The quality of life of people with epilepsy in high-income countries is better than in low and middle-income countries. Higher levels of absolute poverty, limited access to health care and medications, a shortage of specialized healthcare workers; increased perceived stigma toward PWE; drug availability, and employment make the quality of life worse for patients in low and middle-income countries [22, 23].

Epilepsy is strongly associated with impaired quality of life. Peoples with epilepsy have a poor quality of life than both the general population and many other chronic diseases. Despite this, attention is not given to the quality of life of people with epilepsy other than targeting symptom reduction [3]. People with epilepsy had been stigmatized and reduced life changes in every aspect of livelihood which hampered patients' quality of life [24].

In Ethiopia, the prevalence of epilepsy was reported as 5.2/1000 population which is the highest prevalence for ages 10–19 years [25]. The annual incidence of epilepsy in Ethiopia was 64 in 100,000 inhabitants at risk. In addition to this higher prevalence, PWE also was not well treated which was only 1.6% in rural and 13% in urban take the antiepileptic drug the remaining was treated with local herbs, holy water, and amulet [26].

Another study in a rural part of Ethiopia revealed, 60% of patients with epilepsy face different social, psychological, and physical health problems which cause patients to develop a poor quality of life [27]. Although World Health Organization (WHO) call all countries to take action to reduce its burden and improve the quality of life (QOL) of Patients, in many countries including Ethiopia due to limited health care staff; limited health care system; inadequacy of medicine; societal ignorance and misconception and extreme poverty quality of life of PWE is declined [6, 28].

In low and middle-income countries including Ethiopia people with epilepsy and their families suffer from stigma and discrimination. Since peoples with epilepsy are highly marginalized, the quality of their life was not well addressed and few studies were conducted in the region and there is no single study in the study area. Therefore this study was aimed to address the Quality of life and its associated factors among patients with epilepsy at specialized hospitals, Northwest Ethiopia 2019.

## Methods

### Study area and period

The study was conducted in East Gojjam Zone hospitals; Debre Markos Referral and Shegaw Motta district hospital from the 1st March to the 1st April 2019. Debre Markos town is located in northwestern Ethiopia, in Amhara National Regional State, East Gojjam zone, at a distance of 300 km from Addis Ababa, and 265 km from Bahir Dar, the regional capital. According to a national census conducted by the central statistical agency of Ethiopia, in 2012 had an estimated population of 262,497 of whom 129,921 were males. Debre Markos referral Hospital is established in 1957 E.C which is currently giving services to more than 3.5 million population per year in its catchments area with both inpatients and outpatient services. Motta is found in

Northwest Ethiopia, in Amhara Regional state East Gojjam Zone, at a distance of 370 km from Addis Ababa and 120 from Bahir Dar regional capital. Based on the figure from the central statistics agency in 2005, this town had an estimated total population of 31,483, of whom 15,619 were males.

**Study design.** Institution-based Cross-sectional study design was conducted.

## Population

**Source population.** All epilepsy patients who have followed up in Debre Markos Referral Hospital (DMRH) and Motta district hospital were a source population.

**Study population.** All epilepsy patients available during the data collection period of the study on follow-up in Debre Markos Referral and Shegaw Motta District Hospitals.

## Inclusion and exclusion criteria

**Inclusion criteria.** All patients aged 18 years and above with a diagnosis of epilepsy and under treatment with one or more antiepileptic drugs from the outpatient units in Debre Markos Referral (DMRH) and Shegaw Motta District Hospitals (SMDH) for at least 3 months were included.

**Exclusion criteria.** People with epilepsy suddenly develop a loss of consciousness due to a seizure attack at the time of data collection, people with epilepsy unable to communicate due to hearing loss, serious medical conditions, or psychiatric problems, and critically ill were excluded.

## Sample size determination

The sample size for the quality of life was determined using a single population proportion formula, assuming a 95% confidence level and by taking prevalence of population living with epilepsy who had poor quality of life was 45.8% [29] and considering 10% non-response rate. The final sample size was 419.

The sample size determination using factors associated with the quality of life of individuals with epilepsy was calculated by Epi Info 7 Stat Calc program, 2020 using the assumptions. The sample size calculated for the first objective was greater than the sample size determined for the second objective. So, by taking the larger number, the final sample size for the study was 419.

## Sampling technique and procedure

A systematic random sampling technique was used to select the representative sample size of epilepsy from the two hospitals' OPD follow-up clinics. The total number of epilepsy patients from the two referral hospitals was 1,395, in Debre Markos referral hospital (864) and Shegaw Motta district hospital (531). K interval was calculated by dividing the total study population by the sample size which was 5. The sample size was proportionally allocated, 260 from Debre Markos referral hospital and 159 from Shegaw Motta district hospital were selected every five patients based on their order of visit to OPD.

## Data collection tool and procedure

Data was collected by using structured questionnaires regarding socio-demographic characteristics and clinical factors of epilepsy. Moreover, the health anxiety and depression scale is a 14-item questionnaire, commonly used to screen for symptoms of anxiety and depression [30]; the perceived stigma tool to measure the feeling of internalized stigma from the Kilifi

epilepsy stigma scale was adopted from Kenya with a total score of 15 items [31], and Morisky Medication Adherence Scales (MMAS-8) were used to measure antiepileptic drug adherence of patients with epilepsy [14].

The quality of life was assessed using the WHOQOL-BREF questionnaires tool which contains a total of 26 items and a sound, cross-culturally valid assessment of quality of life measuring tool, consisting of four domains: physical health (7 items), psychological health (6 items), social relationships (3 items), and environmental health (8 items); it also contains the first two questions on the general perception of life and health which is scored separately as a benchmark by using a scale [32].

## Study variables

**Dependent variable.** Quality of life of epileptic patients.

**Independent variables. Socio-demographic status**: sex, age, religion, place of residence, marital status, family size, education, income.

*Psycho-social factor.*—perceived stigma.

*Clinical factors.* Duration of illness, frequency seizure, drug type, AED, Drug Duration, drug adherence, and Co-morbid factor anxiety and depression.

## Data quality control

One-day training was given for the data collectors and supervisors to ensure the quality of data. Before the actual data collection, a pre-test was conducted on 21 individuals (5%) using a structured questionnaire. Based on the finding necessary correction was made. The principal investigator and supervisors were actively involved in the supervision of the data collection. The filled questionnaires were checked daily for completeness by the supervisors and principal investigator.

## Operational definitions

**Quality of life.** Quality of life was assessed using the WHOQOL-BREF tool. Categorization was done using the mean scores. Those respondents who scored greater than or equal to the mean were categorized as having **a good** quality of life in WHOQOL-BREF, and those subjects with values less than the mean were categorized as having **poor** quality of life [8].

**Anxiety and depression.** Anxiety and depression were assessed using the HADS scale which was validated in Ethiopia. The scale was used a cut-off score for anxiety and depression of greater than or equal to 8. Those who had scored 8 and greater than 8 has to be categorized to have anxiety and depression those respondents who were scored less than 8 had to be categorized as not having anxiety and depressed [8].

## Data processing and analysis

Data were coded and entered into EPI data version 3.31 and then exported to SPSS version 25 for analysis. Data cleaning was performed by running the frequency of each variable to check the accuracy, inconsistency, and missed value of the data. Descriptive statistics were done and summarized using texts, tables, and graphs based on the type of variables.

Stepwise forward logistic regression was carried out, and in bivariable logistic regression analysis variables having P-value $\leq 0.25$ was a potential candidate for multivariable logistic regression analysis. Model fitness was checked by Hosmer and Lemeshow's Goodness of fit test (p-value = 0.847). The degree of association between independent and dependent variables

was assessed by using an odds ratio with a 95% confidence interval and The level of significance was declared as P value <0.05.

## Ethics approval and consent to participate

The study protocol was approved by the Institutional Review Board (IRB) of the College of Health Sciences, Debre Markos University with IRB number HSC/1024/16/11. Permission was obtained in the form of written informed consent from the study participants and another concerned body of the hospital administers. To ensure confidentiality, any identifying information about the study participants was not indicated on the questionnaires and they were informed that the collected data is used only for research purposes.

## Results

### Socio-demographic characteristics of participants

A total of 402 were interviewed with a response rate of 96%. From the total study participants, 243(60.4%) were male and nearly one-third (37.3%) of the respondents were found in the age group between 25–34 years with the median age of 28 years old. The majority of the study participants (80.8%), were orthodox Christian followers and nearly half of the study participants (47.3%) were married. From the total study subjects more than three-fourth (70.9%) had a family size of 1–3 (Table 1).

### Clinical related factors

Regarding clinical factors, more than one-third (39.8%) of the study participants had a duration of illness up to five years, followed by 103 (35.6%) who were eleven years and above. According to the frequency of seizure 129 (32.1% had seizure-free per one year and followed by 107(26.6%) had one or more seizure attacks per month. More than half (54%) of the study subjects had taken medication less than 5 years duration, followed by 111 (27.6%) who had taken medication for 6–10 years duration.

From the total respondents, three-fourth (74.6%) of them were on monotherapy (single antiepileptic drugs). From those, respondents who had taken medication 170 (42.3%) reported as they had drug adverse effects. Regarding drug adherence status more than half (53.2%) had low adherence followed (37.6%) medium drug adherence. Twenty-one (5.2%) of the respondents had reported that; they had perceived stigmatized by other people because of their illness (Table 2).

Regarding drug adherence status, 177 (44%) had difficulty forgetting pills to take; followed by 145 (36.1%) who had reasonably missed their drug to take. Regarding the frequency of difficulty of remembering drugs they take; more than half of them (47.8%) never frequently forgot their pills whereas 102(25.4%) sometimes forgot their pills (Table 3).

According to the WHO QOL-BREF measurement, nearly half (47.8%) had poor quality of life. The mean (SD) total score on the WHOQOL-BREF scale score was 53.47±18.42 and ranges between minimum values of 4.75 to a maximum value of 95.5. The WHOQOL BREF also covers four different domains of quality of life, physical, psychological, social, and environmental which are shown below (Table 4).

In this study, half of (49.8%) the study participants had poor quality of life from the environmental domain and relatively low scores of poor quality of life were seen in the social domain which was 155 (38.60%). However, the other two domains (physical and psychological domains) had similar frequency distribution of poor quality of life which accounts for 182 (45.30%) (Fig 1).

**Table 1. Distribution of participants by socio-demographic characteristics at Debre Markos and Motta Hospital, East Gojjam Zone, 2019 (n = 402).**

| Variable | Category | Number | Percent (%) |
|---|---|---|---|
| Sex | Male | 243 | 60.4 |
| | Female | 159 | 39.6 |
| Age | 18–24 years | 131 | 32.6 |
| | 25–34 years | 150 | 37.3 |
| | 35–44 years | 92 | 22.9 |
| | 45 years &above | 29 | 7.2 |
| Religion | Orthodox | 325 | 80.8 |
| | Muslim | 73 | 18.2 |
| | Protestant | 4 | 1 |
| Residence | Rural | 205 | 51 |
| | Urban | 197 | 49 |
| Marital status | Married | 190 | 47.3 |
| | Single | 174 | 43.3 |
| | Divorce/widowed | 38 | 9.4 |
| Family size | 1–3 | 285 | 70.9 |
| | 4–6 | 105 | 25.4 |
| | 7 &above | 15 | 3.7 |
| Educational status | Unable to read & write | 114 | 28.4 |
| | Able to read and write | 63 | 15.7 |
| | Primary school | 66 | 16.4 |
| | Secondary school | 66 | 16.4 |
| | Diploma and above | 93 | 23.1 |
| Occupational status | Employed | 76 | 18.9 |
| | Unemployed | 326 | 81.1 |
| Income status | <500 birr | 116 | 28.9 |
| | 500–1000 birr | 84 | 20.9 |
| | 1001–1500 birr | 42 | 10.4 |
| | >1500 birr | 160 | 39.8 |

## Factors associated with quality of life

The result of multivariable logistic regression revealed that older age, lower educational level, those who had low drug adherence, comorbid anxiety, and depression were significantly associated with poor quality of life. Being in the mid-age 25–34 years were nearly three times less likely to have (AOR = 0.36, 95% CI: 0.19, 0.70) poor quality of life than the age group of 18–24 years old. Being unable to read and write was 2.51 times (AOR = 2.51, 95%CI: 1.19, 5.28), and being able to read and write were three times (AOR = 3.11, 95%CI: 1.35, 7.15), more likely to have a poor quality of life as compared to respondents with an education level of diploma and above respectively. Similarly, patients who had medium and low drug adherence eight times (AOR = 8.36, 95%CI: (2.41, 28.98) and fourteen times (AOR = 14.65, 95% CI: 4.35, 49.38) were more likely to have a poor quality of life than high drug adherence respectively. Respondents who had anxiety and depression were nearly four times (AOR = 3.63, 95%CI: 2.55, 8.42) and (AOR = 3.85, 95% CI: 2.16, 6.82) more likely to have a poor quality of life than those who had no anxiety and depression respectively (Table 5).

**Table 2. Distribution of participants by clinical factors at Debremarkos and Motta Hospital, East Gojjam Zone, 2019 (n = 402).**

| Variables | Category | Frequency | Percent (%) |
|---|---|---|---|
| Duration of illness | ≤5 years | 160 | 39.8 |
| | 6–10 years | 99 | 24.6 |
| | ≥11 years | 143 | 35.6 |
| Frequency of seizure | Seizure free for 1year | 129 | 32.1 |
| | ≥1/month | 107 | 26.6 |
| | 1-3/year | 112 | 27.9 |
| | 4-11/year | 54 | 13.4 |
| Medication duration | ≤5 years | 217 | 54.0 |
| | 6–10 years | 111 | 27.6 |
| | ≥11 years | 74 | 18.4 |
| Types of drugs | Monotherapy | 300 | 74.6 |
| | Polytherapy | 102 | 25.4 |
| An adverse effect of drugs | No | 232 | 57.7 |
| | Yes | 170 | 42.3 |
| Drug adherence status | High adherence | 37 | 9.2 |
| | Medium adherence | 151 | 37.6 |
| | Low adherence | 214 | 53.2 |
| Perceived Stigma | No | 381 | 94.8 |
| | Yes | 21 | 5.2 |
| Anxiety status | No | 238 | 59.2 |
| | Yes | 164 | 40.8 |
| Depression status | No | 193 | 48.0 |
| | Yes | 209 | 52.0 |

**Table 3. Distribution of participants' response on drug adherence at Debremarkos and Motta Hospital, East Gojjam Zone, 2019 (n = 402).**

| variables | Category | Frequency | Percent |
|---|---|---|---|
| Forget pills to take | No | 225 | 56 |
| | Yes | 177 | 44 |
| Missing drug for a reason | No | 257 | 63.9 |
| | Yes | 145 | 36.1 |
| Stop drug without doctors permission | No | 317 | 78.9 |
| | Yes | 85 | 21.1 |
| Forget pill when leaving home | No | 253 | 62.9 |
| | Yes | 149 | 37.1 |
| Take all medicine yesterday | No | 99 | 24.6 |
| | Yes | 303 | 75.4 |
| Stop treatment when symptom controlled | No | 305 | 75.9 |
| | Yes | 97 | 24.1 |
| Unable stick your treatment | No | 261 | 64.9 |
| | Yes | 141 | 35.1 |
| Difficulty of remembering | Never | 192 | 47.8 |
| | Once in a while | 92 | 22.9 |
| | Sometimes | 102 | 25.4 |
| | Usually | 11 | 2.7 |
| | All the time | 5 | 1.2 |

**Table 4. Distribution of WHOQOL BREF domains of the respondents at Debre Markos and Motta Hospital, East Gojjam Zone, 2019 (n = 402).**

| Variable | Mean ± SD | Poor QOL frequency | Good QOL frequency |
|---|---|---|---|
| Physical domain | 51.33± 17.07 | 182(45.3%) | 220(54.7%) |
| Psychological domain | 54.16 ±20.70 | 182(45.3%) | 220(54.7%) |
| Social domain | 53.89± 23.95 | 155(38.6%) | 247(61.4%) |
| Environmental domain | 54.47± 20.42 | 200(49.8%) | 202(52.2%) |

Abbreviations:—SD = standard deviation, QOL = quality of life).

## Discussion

In this study, epilepsy affects the quality of life of people living with epilepsy in about one in two patients. The result of this study was revealed that 47.8% (95% CI: 42%, 52%) of the respondents had poor quality of life. This finding is comparable to the studies done on the quality of life of people with epilepsy in Bhutanese (48.8%), Kenya (49%), and Addis Ababa Ethiopian (45.8%) [32, 33]. This might be due to using the same standardized tools and similar cut-off points to categorize the outcome of interest. The other possible justification for this similarity might be those studies were conducted in developing countries.

The finding of this study was higher than the studies which were conducted in Taiwan (33.29%), Brazil (31.27%), and Colombia (30%) in which most of the respondents had a good quality of life [19, 34–36]. This difference might be due to quality treatment in holistic approach than the traditional approach and higher living standard in these developed countries as well as with the economic and financial barriers to the availability of treatment in developing countries as well as attitude change across countries.

In the current study, the poor quality of life in the physical domains (45.3%), psychological domain (45.3%), and environmental domain (49.8%) had higher than the social domain

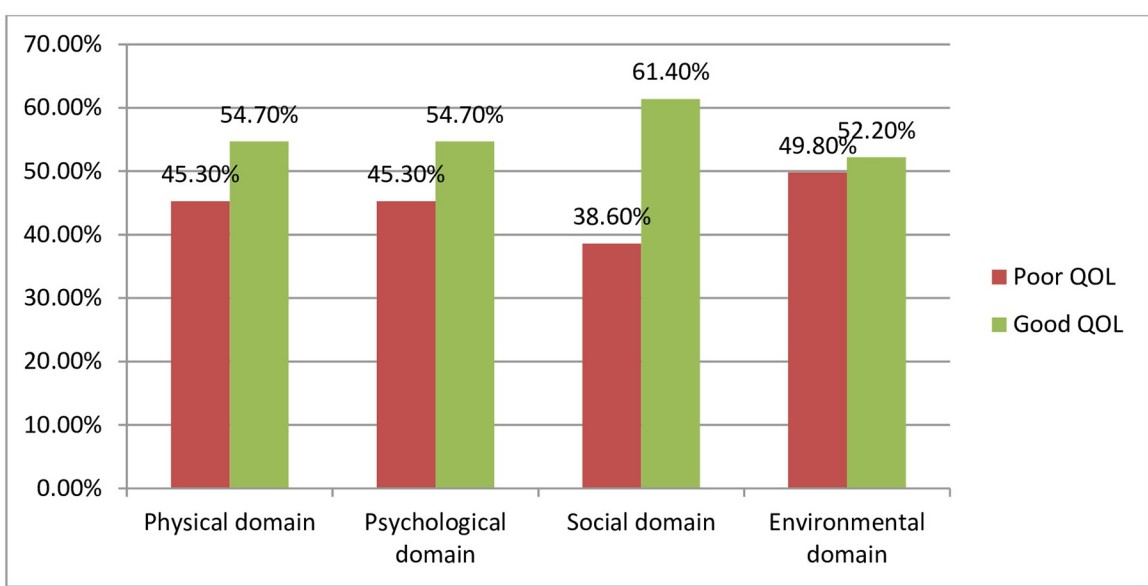

**Fig 1. WHO QOL BREF domains and their frequency among epileptic patients attending at Debre Markos and Motta hospital, 2019 (n = 402).**

**Table 5. Bivariate and Multivariable analysis of variables associated with quality of life among epileptic patients at Debre Markos and Motta hospital, East Gojjam zone, Ethiopia, 2019 (n = 402).**

| Variables | | Quality of life | | | |
|---|---|---|---|---|---|
| | | poor | Good | COR (95% CI) | AOR (95% CI) |
| Sex | Male | 104 | 139 | 1 | 1 |
| | Female | 88 | 71 | 1.66 (1.11,2.48) | 1.25 (0.73, 2.12) |
| Family size | 1–3 | 130 | 155 | 1 | 1 |
| | 4–6 | 57 | 45 | 1.51(0.96, 2.38) | 1.18 (0.61,2.30) |
| | > = 7 | 5 | 10 | 0.60(0.20,1.79) | 0.51 (0.11, 2.37) |
| Occupation | Employed | 25 | 51 | 1 | 1 |
| | Unemployed | 167 | 159 | 2.14(1.27, 3.62) | 0.88 (0.34, 2.26) |
| Income | <500birr | 62 | 54 | 1.64(1.01, 2.65) | 1.24 (0.52, 2.96) |
| | 500–1000 birr | 43 | 41 | 1.49 (0.88, 2.54) | 1.01(0.48, 2.14) |
| | 1000–1500 birr | 21 | 21 | 1.42(0.72, 2.82) | 0.79(0.32, 1.98) |
| | >1500birr | 66 | 94 | 1 | 1 |
| Seizure | Seizure free for 1 year | 49 | 80 | 1 | 1 |
| | > = 1/month | 60 | 47 | 2.08(1.24, 3.51) | 1.52 (0.77, 3.04) |
| | 1-3/year | 52 | 60 | 1.42 (0.85, 2.37) | 0.99(0.51,1.93) |
| | 4-11/year | 31 | 23 | 2.20(1.15, 4.20) | 1.50(0.64, 3.49) |
| Adverse effect | No | 99 | 133 | 1 | 1 |
| | Yes | 93 | 77 | 1.62(1.09,2.42) | 1.37(0.81, 2.32) |
| Duration of illness | ≤5 years | 76 | 84 | 1 | 1 |
| | 6–10 years | 39 | 60 | 0.72 (0.73, 1.20) | 0.58 (0.30,1.12) |
| | ≥11 years | 77 | 66 | 1.29 (0.82, 2.03) | 1.68 (0.92,3.08) |
| Age | 18–24 years | 66 | 65 | 1 | 1 |
| | **25–34 years** | **65** | **85** | **0.75 (0.47, 1.21)** | **0.36(0.19, 0.70)** |
| | 35–44 years | 45 | 47 | 0.94 (0.55, 1.61) | 0.49(0.23, 1.06) |
| | 45years & above | 16 | 13 | 1.21(0.54, 2.72) | 0.73 (0.24, 2.27) |
| Educational status | **Unable to read & write** | **68** | **46** | **3.61(2.02, 6.48)** | **2.51(1.19, 5.28)** |
| | **Able to read and write** | **34** | **29** | **2.87 (1.47, 5.59)** | **2.31 (1.35,7.15)** |
| | Primary school | 33 | 33 | 2.44 (1.27, 4.72) | 1.04 (0.45, 2.43) |
| | Secondary school | 30 | 36 | 2.04(1.05, 3.94) | 1.48 (0.64,3.42) |
| | Diploma and above | 31 | 99 | 1 | 1 |
| | **Medium** | **53** | **98** | **4.46 (1.50, 13.27)** | **8.36 (2.41,28.98)** |
| | **Low** | **135** | **79** | **14.10 (4.82, 41.27)** | **14.65(4.35,49.38)** |
| Anxiety | No | 68 | 170 | 1 | 1 |
| | **Yes** | **124** | **40** | **7.75 (4.92, 12.21)** | **3.63 (2.55, 8.42)** |
| Depression | No | 45 | 148 | 1 | 1 |
| | **Yes** | **147** | **62** | **7.80 (4.99, 12.19)** | **3.85 (2.16,6.82)** |
| Perceived stigma | No | 174 | 207 | 1 | 1 |
| | Yes | 18 | 3 | 7.14 (2.07, 24.64) | 3.13(0.69, 14.19) |

(38.6%). Whereas a study which was conducted in Brazil showed that the physical domain (27.6%), psychological domain (33.3%), and social domain (32.1%) were found to be a higher poor quality of life than the environmental domain (25.0%) [34, 35].

This difference in the two domains might be, in the current study personal relationship, social support, and sexual activity was not well addressed by a health care provider and traditional attitude towards the problem might not be disclosed clearly. However, financial

resources, safety, physical environment, and quality health care were better practiced in Brazil and safe for patients with epilepsy.

The present study revealed that the quality of life of patients with epilepsy was significantly associated with age, educational status, drug adherence, comorbid anxiety, and depression.

In this study age group, 25–34 years were nearly three times less likely to have a poor quality of life than their counterparts. This was supported by studies that were conducted in the United States of America and in Jimma teaching hospital [14, 32]. The possible reason for this might be, being that young adults have self-reliance, and they would be physically and mentally active than their counterparts. Whereas an increased age had poor quality of life to people with epilepsy; this might be because of decreased competitiveness to productivity and increased dependency due to age-related physiological change which ends up with the poor quality of life.

The finding of this study showed that educational status was significantly associated with the quality of life of PWE. Those respondents who were unable to read and write were nearly three times more likely to have a poor quality of life than people who had a higher educational level. Respondents who were able to read and write were nearly two times more likely to have a poor quality of life than their counterparts. This finding was in line with studies in the United States of America, India, Kenya, and Ethiopia at Amanuel mental specialty clinic [31, 32, 33, 37, 38]. The possible reason might be lower educational status had more prone to traditional attitude and decreased self-esteem which leads to psychologically unstable about the disease than well educated.

In this study, clinical factors such as anxiety and depression had significantly associated with poor quality of life. Those respondents who had anxiety were nearly four times more likely to have a poor quality of life of people with epilepsy than those respondents who had no anxiety. Respondents who had depression were four times more likely to have a poor quality of life of people with epilepsy than their counterparts. This result was consistent with the studies which were done in Saint Amanuel Mental Specialized Hospital, Addis Ababa, Ethiopia, Northern Taiwan, Poland, Colombia, and Japan Serbia [3, 7, 29, 33, 34, 39–42]. The possible reason why anxiety and depression are associated with poor quality of life in patients with epilepsy could be anxiety and depression were poorly identified in people with epilepsy and the treatment approach might not be holistic which focuses only on what they had currently been diagnosed and treated. Therefore; having this anxiety and depression can have a great impact on the physico-social quality of life of people with epilepsy.

In this study, drug adherence was significantly associated with the poor quality of life of people with epilepsy. Those respondents who had low drug adherence were fourteen times more likely to have a poor quality of life than those respondents who had high drug adherence. Respondents who had medium drug adherence were eight times more likely to have a poor quality of life than their counterparts. This finding was supported by a study which was conducted in Saint Amanuel Mental Specialized Hospital, Addis Ababa [3]. The justification would be the brain needs a constant supply of seizure medicine to work to stop and prevent seizures. Therefore if the drug adherence is low seizures would not be controlled and the quality of life of people with epilepsy might be compromised.

## Conclusion

The result of this study revealed that about one in two epileptic patients had poor quality of life. Age 25–34 years old, low level of education, low drug adherence, and comorbid anxiety and depression, were significantly associated with poor quality of life. Therefore, health care professionals and other concerned health sectors including health service managers should not

only focus on the diagnosis and treatment of the disease but also focus to provide holistic patients care service which is faced to achieve a good quality of life for patients with epilepsy.

## Supporting information

**S1 File. Data collection tool.**
(DOCX)

**S2 File. The dataset used for this study.**
(SAV)

## Author Contributions

**Conceptualization:** Fentahun Minwuyelet, Henok Mulugeta, Benalfew lake, Asmare Getie, Baye Tsegaye, Getachew Mullu.

**Data curation:** Fentahun Minwuyelet.

**Formal analysis:** Fentahun Minwuyelet, Henok Mulugeta, Benalfew lake, Asmare Getie, Baye Tsegaye, Getachew Mullu.

**Funding acquisition:** Fentahun Minwuyelet, Baye Tsegaye, Getachew Mullu.

**Investigation:** Fentahun Minwuyelet, Henok Mulugeta, Dejene Tsegaye, Benalfew lake, Asmare Getie, Getachew Mullu.

**Methodology:** Fentahun Minwuyelet, Henok Mulugeta, Dejene Tsegaye, Asmare Getie, Baye Tsegaye, Getachew Mullu.

**Project administration:** Fentahun Minwuyelet, Getachew Mullu.

**Resources:** Benalfew lake, Baye Tsegaye.

**Software:** Fentahun Minwuyelet, Dejene Tsegaye, Asmare Getie, Baye Tsegaye, Getachew Mullu.

**Supervision:** Fentahun Minwuyelet, Dejene Tsegaye, Asmare Getie, Baye Tsegaye, Getachew Mullu.

**Validation:** Fentahun Minwuyelet, Getachew Mullu.

**Visualization:** Benalfew lake, Asmare Getie.

**Writing – original draft:** Fentahun Minwuyelet.

**Writing – review & editing:** Fentahun Minwuyelet, Asmare Getie, Getachew Mullu.

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
