## [Decision Letter · Decision Letter 0]

1 Nov 2021

PONE-D-21-32281Quality of life and associated factors among patients with epilepsy at specialized hospitals, Northwest Ethiopia; 2019PLOS ONE

Dear Asmare Getie

Thank you for submitting your manuscript to PLOS ONE. After careful consideration, we feel that it has merit but does not fully meet PLOS ONE’s publication criteria as it currently stands. Therefore, we invite you to submit a revised version of the manuscript that addresses the points raised during the review process.

We look forward to receiving your revised manuscript.

Kind regards,

Muhammad Junaid Farrukh

Academic Editor

PLOS ONE

Journal Requirements:

Additional Editor Comments:

Dear Author, after carefully reviewing your manuscript and reviewers comments, this manuscript is not suitable to publish in this state and require major revision. Please do the changes highlighted by the reviewers and submit us again.

Reviewers' comments:

Reviewer's Responses to Questions

**Comments to the Author**

1. Is the manuscript technically sound, and do the data support the conclusions?

Reviewer #1: Yes

Reviewer #2: Partly

2. Has the statistical analysis been performed appropriately and rigorously? 

Reviewer #1: Yes

Reviewer #2: Yes

3. Have the authors made all data underlying the findings in their manuscript fully available?

Reviewer #1: Yes

Reviewer #2: No

4. Is the manuscript presented in an intelligible fashion and written in standard English?

Reviewer #1: No

Reviewer #2: Yes

5. Review Comments to the Author

Reviewer #1: Overall interesting study and results presented

English editing required for changes for grammatical coherence throughout the text in addition to correcting typographical errors.

Introduction

Overall a few changes need to be made for grammar and for it to read well. Examples are:

Line 52 kindly replace ITS' with ITS without the apostrophe

Lines 54, 55 I believe that sentence would read better if IS and HAD are replaced with "HAS"

Line 57 replacing increased with increases

Lines 59-62 Restructuring the sentence and possibly dividing into two sentences. If it does not change the thought the authors want to convey, a suggestion is as follows: "Therefore, it is important that health care providers routinely measure the impact of the complex pharmaco–psycho-social therapy given. Assessing the success of such a holistic method of care by determining the extent of seizure control with medication and monitoring for reduction of seizure frequency is important"

Line 76 consider adding "is" after the 80%

Lines 77-81, 82-88, 89-92 Kindly review grammar and structure of sentences Consider breaking into multiple sentences for each if possible.

Kindly state aims and objectives of the study as in abstract.

Methods

Line 120 DMRH has not been defined. This seems to be Debre Markos Referral Hospital. Kindly define this abbreviation before use here. Same for SMDH in line 128

Line 145 Duplication. It is already stated earlier that the study was conducted at two hospitals from line 107-117

Line 182 Data Processing and anlysis.

How was the regression carried out? for example Stepwise forward?

Results

Would have been interesting to see a breakdown of the results based on institutions since the study was institution based. Kindly do so if possible to identify specific patterns for the individual hospitals if any. Nevertheless, the Results are still interesting and well done .

Discussion

English editing needed once again for better coherence and grammar.

Lines 279-282. Sentence not clear. Kindly review. I believe the authors are communicating that in those other studies, respondents had higher quality of life than in their study. This could be more clearly stated.

Lines 295-303 This paragraph may benefit from restructuring again mainly for coherence with grammar.

lines 317-322 Sentences not clear. This is an important finding and the interpretation must be very clear to any reader with no ambiguity.

Lines 329-333 also need more clarity.

Line 346 typo. "physico-social"

Conclusion

Also important to highlight that middle age was associated with higher quality of life. It is also an important positive finding.

List of abbreviations

No CRO in the text. Rather COR in the results section. Kindly amend.

Reviewer #2: Comments

This is a valuable piece of work that examine quality of life for epilepsy patients using WHO HRQoL-BERF. To determine associated factors of HRQoL authors also considered some clinical factors (adherence, anxiety, and depression) and psychological factor like perceived stigma that also measured using some well-known established scales. It is a very good combination for data base research using different scale for epilepsy patient.

However, substantial improvement is required especially in introduction and methodology section. Some suggestions to improve the manuscript are as follows:

Overall comments

1. What is the relevance of this study as of few similar studies published in recent year on the same patients in the same region? More recently (in 2021) published two papers and one of them in PLOSONE as well (DOI: https://doi.org/10.1371/journal.pone.0247336).

2. No clear literature gap stated in the introduction section. Major revision is required to update on literature review specially to find the current status of Ethiopia to find out the existing literature gap.

3. What is the relevance to use of HRQOL-BERF to measure QOL for epilepsy patients, although there existing another tool (Quality of Life in Epilepsy Inventory (QOLIE-31)) to measure QOL especially for epilepsy patients.

4. It is required to take permission from developer to use developed tools for research purpose. No statement found about to take permission to use different tools (like: MMAS-8, KESS-15, DAS-14, WHOQOL-BERF-26) for the current study.

5. IRB statement should be stated with the IRB number for more transparency.

Specific comments:

Introduction:

1. Required to update with recent publication

2. Significance of the study and literature/knowledge gap should be stated clearly and decent way as huge lacking found here.

Methodology:

1. According to the statement (The total number of epilepsy patients from the two referral hospitals was 1,395, in Debre Markos referral hospital 864 and Shegaw Motta district hospital 531) population size of this study was known. So it is possible to calculate sample size from this known population. Although the current procedure is fine but better to go for strait forward method.

2. Procedure of systematic random sampling should be explain in details

3. Please give the reference to use P-value ≤ 0.2 to select covariates for multiple model. Page 10 L-187-188, In bivariable logistic regression analysis variables having P-value ≤ 0.2 was a potential candidate for multivariable logistic regression analysis.

4. Required to explain properly about the Outcome variable. How classified QOL good vs poor? What is the cutoff value? Have any reference?

5. Similarly explanation also required about SCALING, SCORING and CLASSIFICATION with cutoff point and reference/logic for independent variables ADHERENCE, STIGMA, ANXIETY and DEPRESSION.

Results:

1. In table 5 better to report p-value for odds to avoid confliction as higher p-value was used to select variables for multiple regression

2. Very high odds observed for ADHERENCE categories from table 5. Required to crosscheck it.

3. Better to report the result about model fitness under the respective tables

Discussion:

1. ‘Results with statistics’ were repeated in the several parts of the discussion section with the results section. Repetition should be prohibited due to redundancy

2. Major revision is required on the writing style as the writing of the manuscript failed to follow scientific merit due to unnecessarily describe some points with repetition of results that increased the volume

Reference:

1. Reference is required to update. Reference number 3, 9, 29 updated reference is available.

6. PLOS authors have the option to publish the peer review history of their article (what does this mean?). If published, this will include your full peer review and any attached files.

Reviewer #1: No

Reviewer #2: No

---

## [Author Response · Author response to Decision Letter 0]

6 Nov 2021

Author’s Point-by-Point Response to the Reviewer's and Editors Reports

Title: Quality of life and associated factors among patients with epilepsy at specialized hospitals, Northwest Ethiopia; 2019

Corresponding author: Asmare Getie/ asmaregetie2017@gmail.com

PONE-D-21-32281

Journal: PLOS ONE

Point by point response to Reviewers and Editors

First of all, the authors would like to thank PLOSE ONE Journal editors and the respective reviewers for reviewing this manuscript and providing the necessary comments to be corrected. As per the comments given, we have made modifications and presented point by point to each comment. The authors tried to answer all the issues raised by editorial team and reviewers. Please note that we gave the response in blue font color.

Reviewer 1: comments and the response given from the authors.

Comment 1: Results, Would have been interesting to see a breakdown of the results based on institutions since the study was institution based. Kindly do so if possible to identify specific patterns for the individual hospitals if any. Nevertheless, the Results are still interesting and well done.

Response 1: thank you very much for this suggestion. The two institution are located in same region and same zone, the authors believed that there would not be different patterns in this two hospitals. The institution is not considered as a factor to affect the quality of life of patients with epilepsy. That is why the authors didn’t showed the results as breakdown.

General comments and their response: all the comments given in each line by line as well as the grammar part and way of writing style was modified and corrected accordingly. The authors kindly request the reviewer to appreciate the modification from the track change and corrected manuscript.

Reviewer 2: comments and the response given from the authors.

Comment 1: What is the relevance of this study as of few similar studies published in recent year on the same patients in the same region? More recently (in 2021) published two papers and one of them in PLOSONE as well (DOI: https://doi.org/10.1371/journal.pone.0247336).

Response 1: thank you for your suggestion: even if the two studies was conducted in one region, but they were conducted in different zones. The current study was conducted in East Gojjam Amhara region and that one was conducted in North wolo Amhara region. As we have tried to see the gab, this epileptic patients are highly marginalized and it is very sensitive issue especially in developing countries including Ethiopia. There for further community based as well as institutional based study is very crucial. The researchers are strongly believed that the finding of this study is very significant for both the local and international community as well as for those peoples suffered from epilepsy. In addition since epilepsy is very sensitive issue the belief of the individual their perception towards the problem in two study area might be differ.

Comment 2: No clear literature gap stated in the introduction section. Major revision is required to update on literature review specially to find the current status of Ethiopia to find out the existing literature gap.

Response 2: thank you very much for this constructive suggestion, based on your suggestion the introduction part was revised and the existing current status in Ethiopia was addressed. You can kindly check the modification in the clean manuscript and manuscript with track change.

Comment 3: What is the relevance to use of HRQOL-BERF to measure QOL for epilepsy patients, although there existing another tool (Quality of Life in Epilepsy Inventory (QOLIE-31)) to measure QOL especially for epilepsy patients.

Response 3: thank you very much for this nice suggestion. Reliability, content and construct validity testing has been performed on the QOLS and a number of translations have been made throughout different continent of the world. The QOLS is a valid instrument for measuring quality of life across patient groups and cultures and is conceptually distinct from health status or other causal indicators of quality of life and it is very appropriate to use it in the contextualized ways to the population characters of different study areas. Many different researchers have used this standardized, well validated and highly reliable tools. As a researcher we believed that even we have used Quality of Life in Epilepsy Inventory (QOLIE-31)) to measure QOL especially for epilepsy patients.

Comment 4: It is required to take permission from developer to use developed tools for research purpose. No statement found about to take permission to use different tools (like: MMAS-8, KESS-15, DAS-14, WHOQOL-BERF-26) for the current study.

5. IRB statement should be stated with the IRB number for more transparency.

Response 4: thank you very much for this concern. We believed that putting citation for this tool we have used can take over the permission statement, since citation is one way of giving acknowledgment/recognition to the developer of the original work. Citation was putted there.

Comment 5: IRB statement should be stated with the IRB number for more transparency.

Response 5: thank you very much, it was corrected accordingly HSC/1024/16/11

Comment 6: 1. Required to update with recent publication

Response 6: thank you, it was modified accordingly.

Comment 7: Significance of the study and literature/knowledge gap should be stated clearly and decent way as huge lacking found here.

Response 7: thank you very much for this suggestion, based on this suggestion modification was made and the researchers have tried to address the significance of the study and knowledge gap.

Comment 8: According to the statement (The total number of epilepsy patients from the two referral hospitals was 1,395, in Debre Markos referral hospital 864 and Shegaw Motta district hospital 531) population size of this study was known. So it is possible to calculate sample size from this known population. Although the current procedure is fine but better to go for strait forward method.

Response 8: thank you very much, as you have said we might calculate strait forward, but already we have used the presented procedure.

Comment 9: Procedure of systematic random sampling should be explain in details

Response 9: thank you for this suggestion, the procedure was presented in detail, you are kindly appreciate from the modified manuscript or from the track change.

Comment 10. Please give the reference to use P-value ≤ 0.2 to select covariates for multiple model.

Response 10: thank you very much, actually we have used P-value ≤ 0.25 to select covariates for multiple model, it was a type error, and was corrected accordingly.

Comment 11: Required to explain properly about the Outcome variable. How classified QOL good vs poor? What is the cutoff value? Have any reference?

5. Similarly explanation also required about SCALING, SCORING and CLASSIFICATION with cutoff point and reference/logic for independent variables ADHERENCE, STIGMA, ANXIETY and DEPRESSION.

Response 11: to categorize the outcome variable we have used mean. The mean score of each domain and the total score were also calculated since quality of life measures in studies are often presented as means. Therefore, categorization was done using the mean scores of WHOQOL-BREF. Those respondents who scored greater than or equal to mean were categorized as having GOOD QOL in WHOQOL-BREF, and those subjects with values less than the mean, were categorized as having POOR QOL

We have used Health anxiety and depression scale which was a 14-item questionnaire, commonly used to screen for symptoms of anxiety and depression. The 14- items can be separated into two 7-item sub-scales for each anxiety and depression from 0(zero) to 3(three) likert scale. HADS scale was validated in Ethiopia. The scales was used a cut -off score for anxiety and depression of greater than or equal to 8. Those who had scored 8 and greater than 8 has to be categorized to have anxiety and depression with negative scoring method, the higher the score the more have anxiety and depressed. We have used Kilifi epilepsy stigma scale in Kenya score of 15 to asses Stigma and which was validated in Ethiopia with simple three point Likert scale scoring of” not at all”(0), “sometimes”(1) and “always”(2) and we have used 66th percentile as cut of point to say stigma or not stigma.

To asses drug adherence we have used Morisky Medication Adherence Scales (MMAS-8) which was already used in Ethiopia and it was categorized based on that cutoff points.

Comment 12: Very high odds observed for ADHERENCE categories from table 5. Required to crosscheck it.

Response 12: thank you very much, we have checked the table as well as the model and the output is same. This might indicated that adherence is highly significant variable.

Comment 13: Better to report the result about model fitness under the respective tables

Response 13: thank you for this constructive suggestion, a report about model fitness was incorporated on the manuscript. 

Comment 14: ‘Results with statistics’ were repeated in the several parts of the discussion section with the results section. Repetition should be prohibited due to redundancy

Response 14: thank you very much for this critical suggestion, based on the comment we have made a great revision and redundancy was removed. You can kindly appreciate this modification either from the track change or the modified manuscript.

Comment 15: Major revision is required on the writing style as the writing of the manuscript failed to follow scientific merit due to unnecessarily describe some points with repetition of results that increased the volume

Response 15: thank you very much for this constructive comment, based on the suggestion provided we have made a major revision and the writing style as well as the language flow was modified accordingly. You can kindly appreciate this modification either from the track change or the modified manuscript.

Comment 16: Reference is required to update. Reference number 3, 9, 29 updated reference is available.

Response 16: thank you, based on your suggestion given the reference was updated

---

## [Editor Report · Decision Letter 1]

6 Jan 2022

Quality of life and associated factors among patients with epilepsy at specialized hospitals, Northwest Ethiopia; 2019

PONE-D-21-32281R1

Dear Dr. Asmare Getie,

We’re pleased to inform you that your manuscript has been judged scientifically suitable for publication and will be formally accepted for publication once it meets all outstanding technical requirements.

Kind regards,

Muhammad Junaid Farrukh

Academic Editor

PLOS ONE

Additional Editor Comments (optional):

we are pleased to inform you that your revised manuscript is suitable for publication
---

## [Editor Report · Acceptance letter]

17 Jan 2022

PONE-D-21-32281R1

Quality of life and associated factors among patients with epilepsy at specialized hospitals, Northwest Ethiopia; 2019

Dear Dr. Getie:

I'm pleased to inform you that your manuscript has been deemed suitable for publication in PLOS ONE. Congratulations! Your manuscript is now with our production department.

Kind regards,

on behalf of

Dr. Muhammad Junaid Farrukh 

Academic Editor

PLOS ONE